# EFFICIENT NEURAL MACHINE TRANSLATION WITH PRIOR WORD ALIGNMENT

## ABSTRACT

Prior word alignment has been shown indeed helpful for better translation, if such prior is good enough and can be acquired in a convenient way at the same time. Traditionally, word alignment can be learned through statistical machine translation (SMT) models. In this paper, we propose a novel method that infuses prior word alignment information into neural machine translation (NMT) to provide hints or guidelines for the target sentence at running time. To this end, previous works of similar approaches should build dictionaries for specific domains, or constraint the decoding process, or both. While being effective to some extent, these methods may greatly affect decoding speed and hurt translation flexibility and efficiency. Instead, this paper introduces an enhancement learning model, which can learn how to directly replace specific source words with their target counterparts according to prior alignment information. The proposed model is then inserted into a neural MT model and augments MT input with the additional target information from the learning model in an effective and more efficient way. Our novel method achieves BLEU improvements (up to 1.1) over a strong baseline model on English-Korean, English-to-German and English-Romanian translation tasks.

## 1 INTRODUCTION

As neural machine translation (NMT) models have become the dominant approach in the machine translation task, the explicit word alignment model, which is an essential intermediary result from the training of most statistical machine translation (SMT) models (Koehn et al., 2003; Och & Ney, 2004; Ganchev et al., 2008), seems becoming increasingly obsolete. Prior research suggests that the attention mechanism of NMT systems takes over the word alignment model of SMT systems (Bahdanau et al., 2014). However, the word alignment information extracted from the attention mechanism is far from gold alignment and even performs much worse than automatic word aligner such as FastAlign or GIZA++. In this study, we focus on the use of prior word alignment in the NMT system to improve translation performance.

With the guidance of good enough known word alignment, replacing some words in the source sentence with semantically corresponding words in the target language leads to better translation or user-desired translation, and it is also known as a tip to use the translator well. As in Figure 1, we can see that an open translation system [1] generates a better translation closer to the target sentence when some words of the target sentence are provided in the source sentence. In order words, a user can use specific alignment, such as 공개 ↔ *released* and 사진 ↔ *picture*, to get a desired translation. The case in Figure 1 happens because word alignment between source and target sentences more or less holds no matter how the model acquires such alignment. Besides, not all word alignment may help and only those good enough word alignment can truly enhance the model. When the concerned language pair shares a large vocabulary, such good enough alignment may be easily obtained and then conveniently works in the proposed early substituting way. This work will right explore an effective way to figure out those 'good' enough certain alignment for neural machine translation enhancement.

Previous studies of similar approaches can be largely divided into two categories: *constraint decoding* and *augmenting MT input with its corresponding target information*. The former is to leverage

---

[1] https://translate.google.com

pre-specified translations to guide the decoding procedure in the modified NMT architecture, such as an additional attention layer (Alkhouli et al., 2018; Song et al., 2020) and a modification to beam search (Post & Vilar, 2018; Hu et al., 2019). They can be useful in certain applications, where the user wants to enforce specific translation of certain words. However, if they do not use information (e.g., alignment and dictionary) extracted from a ground truth sentence pair, they can instead lead to lower translation quality due to strict enforcement of constraints (Dinu et al., 2019). Also, the constrained decoding methods using a modified beam search cause computational overhead in translation time. The latter is to augment MT input with pre-defined dictionary entries for a specific domain and let NMT models learn, at training time, how to use the corresponding target term when provided in the source sentence (Dinu et al., 2019; Park & Zhao, 2019). Although they may gain small but constant performance improvement, they require a 'suitable' pre-defined dictionary for translation tasks with a set of constraints. Otherwise, the translation performance is significantly degraded because it relies only on a fixed dictionary without considering the context.

This paper focuses on integrating word alignment information into NMT systems in an effective and efficient way, without building a 'suitable' dictionary beforehand. To this end, we exploit the alignment learned by SMT model, insert an Alignment-Based Word Substitution (**ABWS**) model into NMT system, and the goal of the model is to learn how to replace the input word with the target one at training time. Specifically, the ABWS model's input is the source sentence, and the reference of the ABWS model is the modified source sentence, in which some words are replaced with their corresponding target words according to the prior word alignment. Note that our model only requires alignment information during training, so unlike the previously proposed models, there is no need to process the test dataset. At inference time, the final hidden state of the trained ABWS model is integrated into NMT models to provide additional target information for MT input. The benefits of our model are twofold: (1) Different from *constraint decoding*, which add modifications to the decoding algorithm, our method does not cause computational overhead in inference time. (2) Although the *augmenting MT input* requires a 'good' pre-defined dictionary, our proposed method does not need to construct it separately because the ABWS model can efficiently perform the pre-defined dictionary role in our model. Furthermore, several previous studies inject alignment signal directly into an attention head of Transformer for constraint decoding or better alignment extraction, but they do not lower or change the translation performance (Alkhouli et al., 2018; Garg et al., 2019; Song et al., 2020).

To summarize, we make the following contributions. (1) We propose a novel *augmenting MT input* method that leverages only prior word alignment without pre-defined dictionaries to improve translation performance in NMT system. Therefore, prior alignment information from the automatic word aligner can be effectively injected into the NMT system in any bilingual corpus through our method. (2) In our experiment, our model outperforms strong baseline models such as vanilla Transformer, *constraint decoding*, *augmenting MT input* method on Romanian-English, English to German, and Korean-English translation tasks.

## 2 RELATED WORK

Word alignment is no longer an indispensable component in training NMT models, but recently there has been a resurgence of interest in the community to study word alignment for NMT models due to its better interpretability. For example, previous works used word alignment information to interpret the end-to-end NMT system and analyze translation errors or to extract better alignments from the learned NMT models. There have been also several studies that leverage word alignment to guide NMT decoding directly, and especially Alkhouli et al. (2018) described this approach as a new downstream task of leveraging word alignment (*dictionary-guided decoding task*). In this study, we focus on infusing SMT offered word alignment into NMT system to improve translation performance.

Alkhouli et al. (2018) added a special alignment head, which conditions a lexical model on the alignment information, to the multi-head source-to-target attention module of Transformer decoder. The use of a separate alignment model adds significant computational overhead to the decoding process, requiring special handling to optimize speed. And Song et al. (2020) proposed an approach that introduces a dedicated head in the multi-head Transformer architecture to capture external supervision signals. Besides, these two studies constraint the decoding process to correct translation errors with

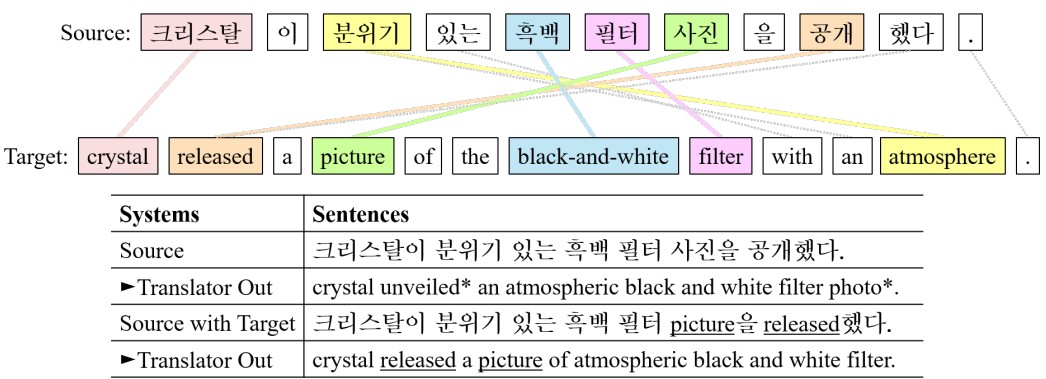

Figure 1: A Korean-to-English translation example of inputting a source sentence, that replaces the input word with its corresponding target one, in an open translation system. The colorful line between the source and target at the top represents 'good' alignment, and the gray dotted line is a trivial one. The underline denotes the target word corresponding to the source one, and the asterisk* is the word to correct using the 'good' alignment.

pre-defined dictionaries and achieve significant improvement on translation tasks. Dinu et al. (2019); Park & Zhao (2019) also used pre-defined dictionaries for a specific domain to let NMT model learn how to use dictionary entries when provided with the input, but they do not constraint the decoding procedure and adopt a non-coercive method that augments MT input with additional information. Furthermore, Dinu et al. (2019) showed that constrained decoding hurt translation performance in their experiments. While our goal resembles that of Dinu et al. (2019); Park & Zhao (2019) (converting MT input to augment it with pre-defined dictionaries for a specific domain), our proposed method does not need to build the pre-defined dictionaries and not limit to a specific domain, all it needs is to feed bilingual word alignment during training.

After NMT has become the dominant MT approach, there have been a variety of studies that integrate external resources (e.g., lemmas, POS tags, named entity, and other linguistic features) into NMT systems. Sennrich & Haddow (2016) proposed a simple but novel method that augments input embedding with its corresponding linguistic features through concatenation operation. Li et al. (2020) presented three ways to integrate the inputs' compressed sentence into NMT systems. Correspondingly, we propose four integration methods to infuse the target information generated from the ABWS model into NMT models.

## 3 MODEL

In this section, we first describe the Transformer (Vaswani et al., 2017) for machine translation. Then we propose our core model, Alignment-Based Word Substitution (**ABWS**) model that replaces part of the source sentence with the corresponding target one according to prior word alignment. Furthermore, we introduce four strategies to fuse ABWS output into NMT system: Source-side Addition (**Src-Add**), Source-side Context Gate (**Src-Gate**), Soruce-side Fusion (**Src-Fusion**) and Target-side Fusion (**Tgt-Fusion**). Figure 2 shows the architecture of our proposed model with Target-side Fusion integration strategy.

### 3.1 TRANSFORMER

A Transformer architecture follows the encoder-decoder paradigm (Sutskever et al., 2014) and has $N$ stacked encoder layers and decoder layers that rely entirely on self-attention networks. A sequence of input words is first fed into a word embedding layer to get word embeddings. Then positional information is injected into the embeddings. The word embeddings are fed into the encoder layer that consists of two sub-modules, namely a self-attention module and a feed-forward module. The self-attention module first creates a query matrix $Q$, a key matrix $K$, and a matrix-vector $V$ from

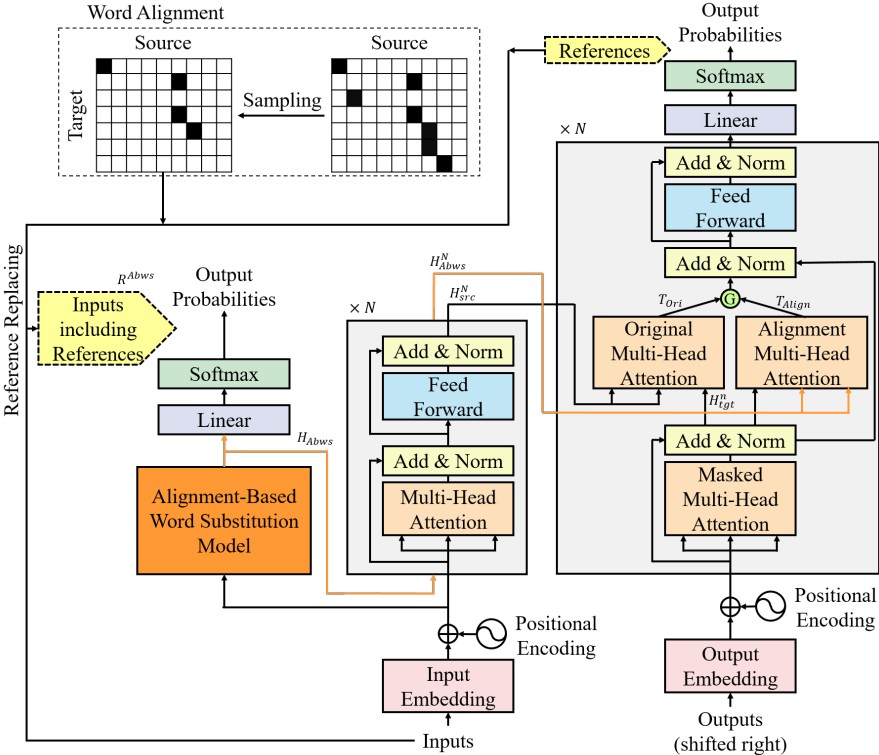

Figure 2: The architecture of our proposed model with Target-side Fusion integration strategy.

each of the word embeddings and product an output matrix as follows:

$$SelfAttn(\boldsymbol{Q}, \boldsymbol{K}, \boldsymbol{V}) = Softmax(\frac{\boldsymbol{QK}}{d_{model}})\boldsymbol{V}, \tag{1}$$

where $d_{model}$ denotes the dimensions of the model. To get a better meaning and context, the Transformer uses different attention heads that are computed parallelly and independently. Multi-head attention is computed from the concatenation of $n$ attention heads output $head_i$:

$$MultiHead(\boldsymbol{Q}, \boldsymbol{K}, \boldsymbol{V}) = Concat(head_1, ..., head_i)\boldsymbol{W^O}, \tag{2}$$

$$head_i = SelfAttn(\boldsymbol{QW_i^Q}, \boldsymbol{KW_i^K}, \boldsymbol{VW_i^V}), \tag{3}$$

where $\boldsymbol{W^O} \in \mathbb{R}^{d \times d}$, $\boldsymbol{W_i^Q}, \boldsymbol{W_i^K}, \boldsymbol{W_i^V} \in \mathbb{R}^{d \times \frac{d}{n}}$ are parameter matrices. The Multi-head attention network is a core one in Transformer. Each encoder layer consists of a self-attention module and a feed-forward module. To preserve auto-regressive property of translation tasks, masked multi-head attention is added to each decoder layer. Finally, a softmax layer based on the decoder's last layer $\boldsymbol{H}_{final}^N$ produces a probability distribution over the target vocabulary:

$$p(\boldsymbol{y}_t | \boldsymbol{y}_1, ..., \boldsymbol{y}_{t-1}, \boldsymbol{x}) = Softmax(\boldsymbol{H}_{final}^N \boldsymbol{W^F}), \tag{4}$$

where $\boldsymbol{W^F}$ is the learned weight matrix, $\boldsymbol{x}$ is the source sentence, and $\boldsymbol{y}_1, \boldsymbol{y}_2, ..., \boldsymbol{y}_t$ is the target words.

### 3.2 ALIGNMENT-BASED WORD SUBSTITUTION MODEL

We propose Alignment-Based Word Substitution (ABWS) model that learns, at training time, how to align the source word with the target one while target words are provided in the source sentence. The model consists of 2 stacked base Transformer's encoder layers and is jointly trained with the NMT model, and the ABWS model input and reference are as follows:

- Input $\boldsymbol{I}_{L \times V}$: Original sentence identical to the input of the NMT model,

- Reference $\boldsymbol{R}_{L \times V}^{Abws}$: Modified sentence that generated by replacing some source words with the target one according to the prior alignment information,

where $L$ is the length of the source sentence and $V$ is the size of the shared vocabulary. At the inference time, the last hidden state of the trained model is incorporated into the input of the NMT model. Formally, one input sentence $\boldsymbol{x} = \{\boldsymbol{x}_1, \boldsymbol{x}_2, \boldsymbol{x}_3, \boldsymbol{x}_4, \boldsymbol{x}_5\}$ is fed into the ABWS model. The model learned to perform suitable replacement with the reference $\boldsymbol{r} = \{\boldsymbol{x}_1, \boldsymbol{y}_5, \boldsymbol{x}_3, \boldsymbol{x}_4, \boldsymbol{y}_1\}$, where two source words $\boldsymbol{x}_2, \boldsymbol{x}_5$ are replaced with its corresponding target word $\boldsymbol{y}_5, \boldsymbol{y}_1$ according to the alignment.

As our heuristics are applied to the use of alignment in the word substitution and model training, we distinguish three cases of bilingual word alignment: one-to-null (unaligned word), one-to-one, and one-to-many (multi-aligned word). For one-to-one alignment, we simply replace the source word with the target one in the sentence. Since unaligned source words are not replaced, this model is simply learning to copy from source word to target one. However, there are some difficulties in processing a multi-aligned word. Unlike normal single-label classification models where class labels are mutually exclusive, the model should be able to classify multi-label data. In other words, if one source word is replaced with multiple target words, this problem cannot be approached with the single-label classification task like the typical sequence generation model. Furthermore, some of the target words may not be the key meaning of the corresponding source one. Therefore, we present three solutions to these issues: (1) Like the noising method of the existing pre-training language models (Devlin et al., 2019; Yang et al., 2019), the replacement process is performed with randomly sampled pairs at a certain ratio from all alignment pairs for each batch. The ratio p for sampling alignment pairs will be empirically determined. (2) It gives the model the flexibility to either learn core meaning from aligned target words or to deviate from fixed source-target word substitution pairs. The parameter of the output linear layer and the parameter of word embedding are shared, and only the parameters of the model are trained. At the inference time, the final hidden state of the model, not the output generated by the `argmax` operation, is directly integrated into the NMT model. This can preserve information about multi-aligned words. In preliminary experiments, we observed a drop of about 2-3 BLUE score when the output generated from the `argmax` is fed into the NMT model through the embedding layer. (3) Inspired by Garg et al. (2019), we implement multi-label classification for the multi-aligned words. Formally, let $\boldsymbol{O}_{L \times V}^{Abws}$ be the output of the model. We minimize the Kullback-Leibler divergence between $\boldsymbol{R}_{l,v}^{Abws}$ and $\boldsymbol{O}_{l,v}^{Abws}$ which is equivalent to optimizing the following cross-entropy loss $\mathcal{L}_a$:

$$\mathcal{L}_a(\boldsymbol{O}^{Abws}) = \frac{1}{V} \sum_{l=1}^{L} \sum_{v=1}^{V} d_{l,v} \boldsymbol{R}_{l,v}^{Abws} \log(\boldsymbol{O}_{l,v}^{Abws}), \tag{5}$$

where $d_{l,v}$ (duplicate alignment penalty) is the inverse frequency of each target index for each alignment sequence, and the goal of this penalty is to exclude words that are repeated too often but have little meaning (e.g., the, a, and so on) from model training. We train our model to minimize $\mathcal{L}_a$ in conjunction with the standard translation loss $\mathcal{L}_t$. The overall loss $\mathcal{L}$ is:

$$\mathcal{L} = \mathcal{L}_t + \lambda \mathcal{L}_a(\boldsymbol{O}^{Abws}), \tag{6}$$

where $\lambda$ is a hyper-parameter, which is set to 0.1 in our experiments.

### 3.3 INTEGRATION STRATEGIES

While many different ways have been explored to augment MT input with additional information, we consider four strategies to incorporate the last hidden state of the ABWS model into NMT system: **Src-Add**, **Src-Gate**, **Src-Fusion** and **Tgt-Fusion**. Given the last hidden state of the ABWS model $\boldsymbol{H}_{Abws} \in \mathbb{R}^{l \times d}$ and input word embeddings $\boldsymbol{S} \in \mathbb{R}^{l \times d}$, where $l$ is the length of source sentence , these four integration strategies are performed as follows.

**Source-side Addition (Src-Add)** is to add $\boldsymbol{S}$ to $\boldsymbol{H}_{Abws}$:

$$SrcAdd(\boldsymbol{S}, \boldsymbol{H}_{Abws}) = \boldsymbol{S} + \boldsymbol{H}_{Abws}. \tag{7}$$

**Source-side Context Gate (Src-Gate)** is to use a context gate $\boldsymbol{G} \in \mathbb{R}^{l \times d}$ for fusing $\boldsymbol{S}$ and $\boldsymbol{H}_{Abws}$:

$$SrcGate(\boldsymbol{S}, \boldsymbol{H}_{Abws}) = ConGate(\boldsymbol{S}, \boldsymbol{H}_{Abws}) = \boldsymbol{G} \odot \boldsymbol{S} + (1. - \boldsymbol{G}) \odot \boldsymbol{H}_{Abws}, \tag{8}$$

$$\boldsymbol{G} = \sigma(MLP([\boldsymbol{S}; \boldsymbol{H}_{Abws}])), \tag{9}$$

Table 1: Corpora statistics and AER [%] w.r.t FastAlign and GIZA++.

| Corpus | Train | Valid | Test | BPE | FastAlign | GIZA++ |
|---|---|---|---|---|---|---|
| Romanian↔English | 399K | 1999 | 1999 | 50K | 35.4/35.5 | 30.9/31.8 |
| English→German | 1.84M | 2999 | 3003/2196 | 40K | 30.9 | 20.7 |
| Korean↔English | 767K | 2000 | 2000 | 32K/32K | - | - |

where $\sigma$ is the logistic sigmoid function, $\odot$ is the element-wise multiplication, and $[;]$ denotes the concatenation operation.

**Source-side Context Gate** (**Src-Fusion**) is to introduce an additional dedicated multi-head attention layer for integrating $\boldsymbol{H}_{Abws}^N$ into the encoder layer:

$$SrcFusion(\boldsymbol{S}, \boldsymbol{H}_{Abws}) = ConGate(\boldsymbol{H}_{src}^n, \boldsymbol{H}_{Align}^n), \tag{10}$$

$$\boldsymbol{H}_{Align}^n = AlignEncMultiHead(\boldsymbol{S}, \boldsymbol{H}_{Abws}, \boldsymbol{H}_{Abws}), \tag{11}$$

$$\boldsymbol{H}_{src}^n = EncMultiHead(\boldsymbol{S}, \boldsymbol{S}, \boldsymbol{S}), \tag{12}$$

where $SrcFusion$ is fed to the FFN module of the encoder layer, and $AlignEncMultiHead$ is an additional dedicated multi-head attention layer that is identical with the original one.

**Target-side Context Gate** (**Tgt-Fusion**) is to introduce an additional dedicated encoder-decoder multi-head attention layer for integrating $\boldsymbol{H}_{Abws}^N$ into the decoder layer:

$$TgtFusion(\boldsymbol{S}, \boldsymbol{H}_{Abws}) = ConGate(\boldsymbol{T}_{Align}, \boldsymbol{T}_{Ori}), \tag{13}$$

$$\boldsymbol{T}_{Align} = AlignEncDecMultiHead(\boldsymbol{H}_{tgt}^n, \boldsymbol{H}_{Abws}^N, \boldsymbol{H}_{Abws}^N)), \tag{14}$$

$$\boldsymbol{T}_{Ori} = OriEncDecMultiHead(\boldsymbol{H}_{tgt}^n, \boldsymbol{H}_{src}^N, \boldsymbol{H}_{src}^N)), \tag{15}$$

$$\boldsymbol{H}_{Abws}^N = FFN(EncMultiHead(\boldsymbol{H}_{Abws}, \boldsymbol{H}_{Abws}, \boldsymbol{H}_{Abws})), \tag{16}$$

$$\boldsymbol{H}_{src}^N = FFN(EncMultiHead(\boldsymbol{S}, \boldsymbol{S}, \boldsymbol{S})), \tag{17}$$

where $TgtFusion$ is fed to the FFN module of the decoder layer, and $AlignEncDecMultiHead$ is an additional dedicated encoder-decoder multi-head attention layer.

## 4 EXPERIMENTS

To verify that the proposed method is effective, we perform experiments on three language pairs: Romanian-English (EN↔RO), English to German (EN→DE) and Korean-English (KO↔EN). For all translation tasks, we use BLEU (Papineni et al., 2002) for the evaluation of translation quality. All the model training is on a single NVIDIA Tesla V100 GPU.

### 4.1 DATASETS

Training data and test data for EN↔RO translation are Europarl v8 corpus and newstest2016, respectively. For EN→DE, we use the Europarl v7 news datasets as training data, newstest2016 as validation data, and newstest2014 and newstest2015 as test data. In order to evaluate alignment quality on two automatic word aligner, we use the the gold alignments for EN↔RO [2] and EN→DE [3] For KO↔EN translation, we use news dataset that provided by AIHub [4] and split the dataset for the validation and test data. For all dataset, we first tokenize three languages (English, Romanian, German) using Moses (Koehn et al., 2007) and Korean data using KoNLPy [5] toolkit and apply Byte-Pair-Encoding (Sennrich et al., 2016). Table 1 shows the statistics of datasets and alignment error rate (Och & Ney, 2000) (AER) for EN↔RO and EN→DE.

---

[2] http://web.eecs.umich.edu/~mihalcea/ wpt/index.html#resources

[3] https://www-i6.informatik.rwth-aachen.de/goldAlignment/

[4] http://www.aihub.or.kr/

[5] https://konlpy.org/en/latest/

Table 2: Experimental results on three translation task. "+" represent significantly better systems than the corresponding baseline Transformer at a p-value $< 0.05$. Time(s) denotes the average translation time (second) per sentence.

| Systems | RO→EN | | EN→RO | | KO→EN | | EN→KO | |
|---|---|---|---|---|---|---|---|---|
| Transformer | 30.84 | | 32.20 | | 39.50 | | 35.63 | |
| | Fast | GIZA | Fast | GIZA | Fast | GIZA | Fast | GIZA |
| **Src-Add** | 31.26 | 31.33 | 32.55 | **32.60** | 40.37+ | 40.34 | 36.01 | 36.04 |
| **Src-Gate** | 31.17 | 31.29 | 32.21 | 32.25 | 40.46 | 40.42+ | 36.11 | 36.01 |
| **Src-Fusion** | 31.22 | 31.33 | 31.90 | 31.88 | 40.35+ | 40.44+ | 35.91 | 35.03 |
| **Tgt-Fusion** | 31.41 | **31.62+** | 32.44 | 32.46 | **40.63+** | 40.50+ | 36.19+ | **36.21** |

(a) RO↔EN (WMT16) and KO↔EN (AIHub)

| Systems | WMT14 | | Time(s) | WMT15 | | Time(s) | #Params |
|---|---|---|---|---|---|---|---|
| Transformer | 26.87 | | 0.336 | 28.98 | | 0.304 | 64.4M |
| Constr. dec. | 26.47 | | 0.495 | 28.59 | | 0.472 | 64.4M |
| Train-by-rep. | 26.78 | | 0.340 | 29.06 | | 0.314 | 64.4M |
| Train-by-app. | 26.88 | | 0.343 | 28.93 | | 0.319 | 64.4M |
| | Fast | GIZA | Time(s) | Fast | GIZA | Time(s) | #Params |
| **Src-Add** | 26.85 | 26.99 | 0.352 | 29.11 | 29.23 | 0.314 | 70.7M |
| **Src-Gate** | 27.12 | 27.28 | 0.341 | 29.10 | 29.33 | 0.320 | 71.2M |
| **Src-Fusion** | 27.36 | **27.55** | 0.377 | 29.75+ | **29.88+** | 0.349 | 80.1M |
| **Tgt-Fusion** | 26.70 | 27.23 | 0.402 | 28.54 | 29.15 | 0.362 | 80.1M |

(b) EN→DE (WMT14 and WMT 15)

## 4.2 SETUP AND BASELINES

For Bilingual word aligner, we use the MGIZA++ [6] (Gao & Vogel, 2008), a parallel implementation of GIZA++, and FastAlign [7] toolkit with default parameters. We align the bilingual training corpora with FastAlign for all language pairs. For RO↔EN translation task, we additionally use word alignments produced by MGIZA++ to compare the different word aligners. Both FastAlign and GIZA are used with default settings and all the training corpora are in subword format. Furthermore, the pairs of sentences with an error (e.g., zero sentences) are pruned in the process of generating alignment.

In all experiments for this task, we train all models using a `base` Transformer configuration with an embedding size of 512, 6 encoder and decoder layers, 8 attention heads, shared source and target embeddings, the standard `relu` activation function and sinusoidal positional embedding. We train with a batch size of 3500 tokens and use the validation translation loss for early stopping and update parameters every 8 batches. Furthermore, We optimize the model parameters using Adam optimizer with a learning rate 7e-4 $\beta_1 = 0.9$ and $\beta_2 = 0.98$, learning rate warm-up over the first 4000 steps. Additionally, we apply label smoothing with a factor of 0.1. We average over the last 5 checkpoints and run inference with a beam size of 5. All our experiments were performed using the Torch-based toolkit, Fairseq(-py) (Ott et al., 2019).

In our experiment, we use Transformer (Vaswani et al., 2017) as the baseline model for all language pairs. For EN→DE translation, we compare our approach to the following baselines:

- Constraint decoding [8]: A vectorized lexically *constrained decoding* with dynamic beam allocation reported in Post & Vilar (2018); Hu et al. (2019).
- Training-by-replacing: An *augment MT input* method directly replacing the original term with the target one according to a dictionary (Dinu et al., 2019; Park & Zhao, 2019).
- Training-by-appending: An *augment MT input* method appending the target term to its source version according to a pre-defined dictionary (Dinu et al., 2019).

---

[6] https://github.com/moses-smt/mgiza

[7] https://github.com/clab/fast align

[8] https://github.com/pytorch/fairseq/blob/master/examples/constrained_decoding

They all require a pre-defined dictionary, so we extracted the dictionary from the GIZA++ alignment for a fair comparison. In order to avoid spurious matches, we used the following pruning methods: (1) We first removed meaningless word with several POS tags[9] (e.g., auxiliary, determiner, punctuation, stop words and so on). (2) To make an appropriate one-to-one matching set, we counted the occurrence frequency of each target term matched to the source one, and then adopt the occurrence frequency of the top 1 target term if it is more than 90% of the occurrence frequency of the total target term. (3) Finally, we filtered out entries occurring in the top 500 most frequent English words.

## 4.3 EXPERIMENTAL RESULTS

Table 2 shows the BLEU evaluation of our systems. For the experimental results, we made the following observation: (1) For the integration strategy, Most of them outperformed the baseline Transformer. We can see that each integration strategy has a gap in improving translation performance depending on the language pair or translation direction. For example, **Tgt-Fusion** shows relatively high performance on RO↔EN and KO↔EN translation, but **Src-Fusion** is the best on EN→DE translation. (2) For the automatic word aligner, the results showed that word alignment information of GIZA++ yields a better performance improvement than FastAlign on RO↔EN and EN→DE translation. This means that better alignment information can lead to better translation. (3) For other baseline models, which use pre-defined dictionary building with prior word alignment, the performance is degraded or maintained. This means that our model makes good use of the alignment information. (4) The parameters of our proposed model with the four integration strategies increased 6M to 16M over the baseline Transformer. For inference speed, the decoding time of our model increased by only 1.2 times over the baseline Transformer.

Table 3: Performances on translation tasks, where our proposed model uses **Tgt-Fusion** integration strategy and word alignment from GIZA++. The (a) and (b) are translation performances (BLEU score) with different alignment sampling ratios and different ABWS models, respectively. In (a) and (b), the asterisk denotes our proposed ABWS model setting. The (c) is to evaluate our model on low-resource cases.

| SR | BLEU | SR | BLEU | Systems | BLEU(△) | Systems | BLEU | |
|---|---|---|---|---|---|---|---|---|
| | | | | | | | 14 | 15 |
| 0.0 | 30.94 | | | w/o ABWS | 30.84(0) | | | |
| 0.1 | 30.94 | 0.6 | 31.43 | Bi-LSTM | 30.80(-0.04) | Trans. | 11.37 | 13.14 |
| 0.2 | 30.87 | 0.7 | 31.50 | 1 Trans Enc | 31.10(0.26) | S. Add | 11.55 | 13.55 |
| 0.3 | 31.02 | 0.8 | 31.53 | 2 Trans Enc* | 31.62(0.78) | S. Gate | 11.94 | 13.71 |
| 0.4 | 31.05 | 0.9* | 31.62 | 3 Trans Enc | 31.43(0.59) | S. Fus. | 13.25 | 15.03 |
| 0.5 | 31.25 | 1.0 | 31.26 | 4 Trans Enc | 31.49(0.65) | T. Fus. | 12.90 | 14.78 |
| (a) Alignment SR (RO→EN) | | | | (b) ABWS Models (RO→EN) | | (c) Low-Resource (EN→DE) | | |

## 5 ANALYSIS

### 5.1 EVALUATING ALIGNMENT SAMPLING RATIO p

In order to verify the impact of different ratios p for sampling alignment pairs on translation quality, we conducted the corresponding experiments on RO→EN translation task with our proposed model. When the sampling ratio p = 0.0, it means no use alignment information, and when the sampling ratio p = 1.0, it is equivalent to leveraging all word alignment information. Table 3a shows the experimental results. Hence, the alignment sampling ratio is set to 0.9 for the best performance in all our experiments.

### 5.2 WHICH NEURAL NETWORK IS SUITABLE AS THE ABWS MODEL?

We experimented with different neural networks to determine which neural network could represent the replaced word well for RO→EN translation task. As shown in Table 3b, there was little change when using Bi-LSTM as the ABWS model, and the transformer encoder improved the translation

---

[9]https://spacy.io/

Table 4: Translation examples in which the ABWS model's output augments the MT input. The underline denotes source word and its corresponding target one in the ABWS output .

| | |
|---|---|
| Source | 부동산 시장 에서 비 규제 지역 이 주목 을 받고 있다 . |
| ABWS top 1 | real market 에서 비 regulated 지역 이 주목 을 받고 있다 . |
| top 2 | estate 시장 in non-@@ 규제 areas has attention 를 receiving is . |
| top 3 | 부동산 markets at un@@ regulatory area are drawing its drawing are .. " |
| Output | un@@ **regulated** areas are drawing attention in the **real estate market** . |
| Target | quantitative un@@ regulated areas are gaining attention in the real estate market . |
| Source | 박 시장 은 선거 다음 날 부터 업무 에 들어갔다 . |
| ABWS top 1 | 박 mayor 은 election 다음 날 부터 업무 에 들어갔다 . |
| top 2 | park 시장 the elections next day from work on began .. . |
| top 3 | 박@@ mayoral 는 선거 following same starting business in started " |
| Output | **mayor** park began his work the day after the **election** . |
| Target | one day after the election, park began his job . |
| Source | 인권 을 가르@@ 치지 않는 것 도 인권 침해 라는 말 이 있다 . |
| ABWS top 1 | rights 을 가르@@ 치지 않는 것 도 human violations 라는 말 이 있다 . |
| top 2 | human they teach 't does what also rights violation says says this are .. . |
| top 3 | 인권 he teaching hurt do things even 인권 침해 say say is being and |
| Output | it is also said that not teaching **human rights** is a **violation** of **human rights** . |
| Target | some say that not teaching human rights is also a violation of human rights . |

performance. Therefore, we adopted the 2 stacked transformer encoder as our model, considering its size and performance.

### 5.3 EVALUATION OF ALIGNMENT USAGE ON LOW-RESOURCE CASES

We evaluated our proposed model's performance on a low-resource case that might lead to poor word alignment and our models' lower performance. We first sampled 100,000 sentences from the EN→DE data set, and the AER for that small data set was 34.0. Contrary to our expectations, Table 3c shows that our models achieve even more remarkable performance gains (up to 2 BLEU) on the low-resource case. This means that our model is also useful in low-resource cases.

### 5.4 ANALYSIS OF ALIGNMENT USAGE IN TRANSLATION

In this subsection, We observed how the ABWS model utilizes prior word alignment in the NMT model, which applies **Tgt-Fusion** integration strategy on the KO→EN translation task. Table 4 shows some translation examples that include top 3 ABWS output. We can see that the ABWS model replaces some source words with target one and the replaced target words appears in the translation output. Moreover, from the Korean word 시장 (*market, mayor*) in the first and second translation examples, we can see that the ABWS model distinguishes the homophone in each sentence. This means that the model's replacement process takes context into account. Another observation is that the ABWS model learned well for multi-aligned words. For example, in the third example,인권(*human rigths*) is aligned with two target words ('*human*' and '*rights*'), and the ABWS model catches them well.

## 6 CONCLUSION

This work presents a novel solution for the effective and efficient fusion of NMT and SMT. Especially, we explore an efficient way of exploiting prior word alignment offered by SMT models for NMT enhancement during the model training phase instead of constraint decoding in previous work which may slow down the inference. In detail, to augment NMT input, we design an extra model that learns how to replace specific source words with their corresponding target ones according to the prior word alignment. Our method helps yield significant performance improvement compared to a strong baseline on three translation tasks, which verifies the effectiveness of the proposed method.

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
