# OpenReview forum: "Efficient Neural Machine Translation with Prior Word Alignment"
_ICLR.cc/2021/Conference — Reject_

### Official Review · AnonReviewer3 · 2020-10-29
**Interesting problem but the modeling is confusing**

**Rating:** 4
**Confidence:** 4

**Review:**

This paper proposes to integrate word alignment obtained from SMT into an NMT system. This is an exciting topic not only because it can help interpretability, but also because the same mechanism could be used e.g. for imposing a specific terminology in translations, something that was relatively easy to do with SMT but is much harder to achieve with NMT. The proposed method involves computing word-word alignment using existing SMT models (GIZA and FastAlign in the experiments) and integrating that information in the decoder of a transformer-based NMT model. Experiments on English-Romanian and English Korean show small improvement over a standard baseline.

Modeling is described at a high level and would require significant guesswork to re-implement. The core of the method is in the word substitution model (Sec 3.2) and how it is integrated in the decoder (Sec 3.3). Unfortunately neither is described with much detail. For example, there are a dozen operators involved in describing the model in Eqs. 1-17, some of them defined, some of them relatively easily guessable, other entirely unclear.
It would greatly help to clearly define the input and outputs of the model (e.g. vectors of context, matrices of probabilities or activation, appropriate dimensions, etc.)

Regarding evaluation, it is a positive to have results on two language pairs, in both directions. In the English-Romanian pair, GIZA seems to produce slightly better results, although the difference is unlikely to be significant. Still, it is odd that only  FastAlign was used for Korean. Although gains seem fairly consistent, they are also very limited and unlikely to be significant (no significance test seems to have been performed). Also the variability of performance (e.g. due to sampling) is not assessed. Finally, the only comparison is to a straight transformer baseline. None of the methods mentioned in the prior art have been tested.

The analysis of the impact of the alignment on the attention weights in Sec 5.2 is interesting. What is the motivation for using different sentences and epoch #, rather than show how alignment is modeled as training progresses (as claimed in the text) on the same sentence?

Misc:
[Sec 5.1]: why is the sampling rate suddenly called the « compression ratio »?

** UPDATE**
Read the author answers, thanks. There is a significant amount of new material added in response to the reviews, including some interesting new findings (e.g. Tab. 3, 4). Unfortunately that did not help the clarity issue. I did not see a new example in Section 3.2.
Basically, the stronger points of the paper (the results) are stronger, but the weaknesses (clarity, motivation) are not really addressed. The new version uploaded by the authors is an improvement but I still lean towards rejection.

---

> ### Author Response · Authors · 2020-11-21
> **Response to AnonReviewer3**
>
> Thank you for your comprehensive review and suggestions to improve the paper. Below are our responses to your comments:
>
> **Need GIZA++ case for Korean-English translation tasks.**
>
> We added the experiment results, which use alignment from GIZA++ for Korean-English translation tasks, in the new pdf version. Unfortunately, we could not find the gold alignment dataset for Korean-English, so we could not check the dataset's alignment error rate (AER) result.
>
> **Describe the model with much detail.**
>
> To explain the model in more detail, We added an example of the input and output of the model to section 3.2.
>
> **What is the motivation for using different sentences and epoch # in Sec 5.2?**
>
> We thought that it would be better to show multiple sentences because the additional attention head is already well learned at the beginning of learning. However, we removed Section 5.2. Instead, we added an analysis of the translation examples of our model to explain in more detail how our model contributes to the translation quality.

---

### Official Review · AnonReviewer2 · 2020-10-29
**Interesting idea of incorporating word alignment in a model**

**Rating:** 5
**Confidence:** 4

**Review:**

This work proposes to incorporate word alignment information as a word substitution model. Basic idea is to jointly train a separate encoder using a cross entropy loss which predicts a source input sequence with words substituted by aligned target words. The learned representation is combined with a Transformer either by simple summation, gating or joint attention mechanism. Experimental results on Romanian/English and Korean/English tasks show very marginal gains over the baseline Transformer.

# Pros

* It is a very interesting work on combining word alignment information in a model by predicting aligned target words by following alignment links.

* Small gains are observed on Romanian/English and Korean/English.

* The attention weight visualization especially for word substituted encoder presents sharper distribution, which might indicate that useful information might be learned in the attention mechanism.

# Cons

* The gains are very small and thus, I suspect that the difference might be negligible when running statistical significance tests.

* Analysis is a bit weak in that it does not compare  the quality of alignment against the translation qualities.

# Details

This work presents an interesting idea, though, I have a couple of concerns to this submission. Thus, I have some hesitation for full acceptance.

* I'd like to see any tradeoffs of word alignment qualities and translation qualities by randomly distorting word alignment links. This work simply tweak sampling rate to vary the ratio of using word alignment links, not distorting links, and thus, it is not a direct measure to verify the quality tradeoff.

  * Thank you very much for adding the comparison of GIZA and fastalign. However, simply swapping a model for alilgnment does not given us details about the tradeoff of alignment quality and the end-to-end results, since alignment models have different characteristic to capture the correspondence in two langauges, e.g., assuming linearlity and/or fertility. I'd rather like to see a much simpler approach of distroting alignment to avoid influence of the models employed for word alignment.

* Given that non-aligned words are also predicted in the word substituted model in Equation 8, I suspect the model is simply learning to copy from input to output for non-aligned words. I think the loss in Equation 8 should be applied only to those words substituted by alignment links.

  * Thanks for the explanation. I'd like to see ablation studies regarding the loss.

* Please verify the experimental results by statistical significance tests.

  * Thank you for adding the tests.

---

> ### Author Response · Authors · 2020-11-21
> **Response to AnonReviewer2**
>
> Thank you for your comments and suggestions to improve the paper. Below are our responses to your comments:
>
> **Verify the experimental results by statistical significance tests.**
>
> We conducted statistical significance tests on all translation tasks and updated Table 2.
>
> **The loss in Equation 8 should be applied only to those words substituted by alignment links.**
>
> We also thought it would be best to apply the loss function only to those words substituted by alignment links. However, we observed that the method significantly lower the translation quality in a preliminary experiment. As a result of analyzing the alignment-based word substitution (ABWS) model's output, we found that the method leads to many invalid word replacement. Even unaligned words were all replaced with improper target terms. The reason is that the loss for unaligned words is not applied, and the replacement process proceeds without considering the context. Therefore, we applied the loss for unaligned words.
>
> **Show tradeoffs of word alignment qualities and translation qualities by randomly distorting word alignment links.**
>
> We can see this tradeoff through the difference in alignment error rate (AER) w.r.t alignment information extracted from FastAlign and GIZA++. We added the AER results for those two auto aligners to Table 1. The AER gap (4% on average) between the two different prior word alignments in Romanian-English translation tasks is small. However, the gap (10%) in English-to-German translation tasks is relatively large. This aspect is the same in translation quality in Table 2. For example, for Romanian-English, the difference in BLEU between the two aligners is 0.075 on average. However, for English-to-German, the difference is 0.26 on average. This means that alignment quality and translation quality are in a proportional relationship.

---

### Official Review · AnonReviewer1 · 2020-10-29
**An interesting idea but needs more progress**

**Rating:** 3
**Confidence:** 5

**Review:**

The paper presents a strategy to integrate prior word alignments into NMT models. It is not clear the motivation for this in the NMT context, especially why the prior alignments are crucial information that is necessary to be given a-priori to the Transformer. Besides this, the description of the method and the discussion of related work is given, SMT methods are briefly mentioned but the usage of the idea in previous work, also SMT literature is necessary.
The applicability of the method is discussed very generally for NMT, however most languages do not have a one-to-one mapping in words and even in extremely low-resource cases it is difficult to see how the alignment information can be useful except for translating for instance foreign words and named entities (in languages where transliteration is not necessary).
The description of the method seems sound and the experiments are performed for two languages with improvements in BLEU scores. Have the authors analyzed what is exactly being improved in the translations?
The paper is mostly clear but has too many grammatical errors and can benefit from revisions and more editing.

---

> ### Author Response · Authors · 2020-11-21
> **Response to AnonReviewer1**
>
> First, thank you very much for your valuable comments.
>
> **why the prior alignments are crucial information that is necessary to be given a-priori to the Transformer?**
>
> Several previous studies [1,2,3] inject alignment signal directly into an attention head of Transformer for constraint decoding or better alignment extraction, but they do not lower or change the translation performance.
> However, Our purpose is to provide the model hints or guidelines for the target sentence at running time. This approach has proven in many previous papers that it can lead to improved quality of translation. In particular, unlike previous studies that must require a pre-defined dictionary or change the beam search, our model achieves the goal only with prior word alignment. As shown in Table 2 (b), our methods outperform the previous methods.
>
> **Need usage of the idea in previous work and SMT literature.**
>
> We added the new baseline models on English-to-German translation tasks to compare our proposed model with them in Table 2. And we cited SMT literature using the alignment model in our revised version.
>
> **Most languages do not have a one-to-one mapping in words.**
>
> Yes, there are one-to-null (unaligned word) and one-to-many (multi-aligned word) in addition to one-to-one for the bilingual word alignment extracted from automatic word aligner. In addition, when we naïvely extracted the dictionary from the alignment, we observed that there are very few one-to-one matches. Therefore, we propose the ABWS model to proceed with the replacement process that considers the context for translation. We added an analysis of the corresponding replacement process in Section 5.4 and Table 4. Specifically, We let the model is simply learning to copy from input to output for non-aligned words and present solutions for one-to-many matches in Section 3.2.
>
> **About extremely low-resource cases**
>
> We added evaluation of alignment usage on low-resource cases in revised version. We found that our proposed model achieved even more significant improvement on small English-to-German dataset (10M). Please see the corresponding analysis in Section 5.3 and Table 3c.
>
> **Have the authors analyzed what is exactly being improved in the translations?**
>
> We added the analysis on translation examples in revised version. Please see the corresponding analysis in Section 5.4. and Table 4.
>
> [1]Tamer Alkhouli, et al. “On the alignment problem in multi-head attention-based neural machine translation”, ACL, 2018.\
> [2]Sarthak Garg, et al. "Jointly learning to align and translate with transformer models", EMNLP, 2018.\
> [3]Kai Song, et al. "Alignment-enhanced transformer for constraining nmt with pre-specified translations", AAAI, 2020.

---

### Author Response · Authors · 2020-11-21
**Paper Revision**

Dear Reviewers and AC, Thanks so much for your time and the constructive reviews. We have updated our revised paper. The detailed revision includes:

1. revise Section 1 to make our motivation clearer. \
2. update Section 3.2 to explain the model in more detail. \
3. provide alignment error rate (AER) w.r.t FastAlign and GIZA++ alignment on Romanian-English and English-to-German datasets . (see Table 1)\
4. conduct statistical significance tests on all translation tasks. (see Table 2) \
5. For a more detailed evaluation, we add experimental results on the English-to-German dataset in Table2b. Besides, we reported the new baseline systems, including Constraint decoding [1], Training-by-replacing [2], and Training-by-appending [2] on the English-to-German translation task. (see Section 4.2)\
6. add an evaluation of alignment usage on low-resource cases (see Section 5.3 and Table 3c) and the analysis on translation examples. (see Section 5.4 and Table 4)


[1]Matt Post and David Vilar. Fast lexically constrained decoding with dynamic beam allocation for neural machine translation. NAACL, 2018. \
[2]Georgiana Dinu, Prashant Mathur, Marcello Federico, and Yaser Al-Onaizan.  Training neural ma-chine translation to apply terminology constraints. ACL, 2019.

---

### Decision · Program_Chairs · 2021-01-07
**Final Decision**

**Decision:**

Reject

**Comment:**

Two reviewers suggested to reject and the other reviewer also thought it below the threshold.